# Comparative Study of the Reactivity of the Tungsten Oxides WO$_2$ and WO$_3$ with Beryllium at Temperatures up to 1273 K

**Martin Köppen**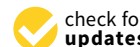

Independent researcher, Gabelsbergerstr. 9, 94032 Passau, Germany; martin_koeppen@gmx.de

**Abstract:** Tungsten oxides play a pivotal role in a variety of modern technologies, e.g., switchable glasses, wastewater treatment, and modern gas sensors. Metallic tungsten is used as armor material, for example in gas turbines as well as future fusion power devices. In the first case, oxides are desired as functional materials; while in the second case, oxides can lead to catastrophic failures, so avoiding the oxidation of tungsten is desired. In both cases, it is crucial to understand the reactivity of tungsten oxides with other chemicals. In this study, the different reactivities of tungsten oxides with the highly-oxophilic beryllium are studied and compared. Tungsten-(IV)-oxide and tungsten-(VI)-oxide layers are prepared on a tungsten substrate. In the next step, a thin film of beryllium is evaporated on the samples. In consecutive steps, the sample is heated in steps of 100 K from room temperature (r. t.) to 1273 K. The chemical composition is investigated after each experimental step by high-resolution X-ray photoelectron spectroscopy (XPS) for all involved core levels as well as the valence band. A model is developed to analyze the chemical reactions after each step. In this study, we find that tungsten trioxide was already reduced by beryllium at r. t. and started to react to form the ternary compounds BeWO$_3$ and BeWO$_4$ at temperatures starting from 673 K. However, tungsten dioxide is resistant to reduction at temperatures of up to 1173 K. In conclusion, we find WO$_2$ to be much more chemically resistant to the reduction agent Be than WO$_3$.

**Keywords:** X-ray photoelectron spectroscopy; physical vapor deposition; X-ray diffraction; tungsten oxide; tungsten dioxide; tungsten trioxide; beryllium; tungstate; tungsten bronze

## 1. Introduction

Tungsten and its oxides play a pivotal role for the solution of current problems in material science. Tungsten oxides have recently received much attention. In consumer electronics, they are used in switchable glass due to their electrochromic properties [1–3]. In wastewater treatment, tungsten oxides are used as catalysts [4]. Modern gas sensors also rely on tungsten oxides [5–7]. In all of these applications, tungsten oxides are the functional material, and a long lifetime of the oxides is desired.

The classical applications of tungsten are in environments with a high heat load, as in turbines or reactors. In these environments, the oxidation of tungsten results in catastrophic events. The reduction of the melting point from approximately 3600 K to approximately 1700 K is severe and eventually leads to the failure of the respective components.

In material research for future fusion reactors, tungsten oxides are adverse in a second way: The formation of tungsten oxide is a severe safety issue. Tungsten trioxide is volatile under the conditions in a fusion reactor. Due to the neutron radiation from the D-T fusion reaction, tungsten can be transmutated

into its radioactive isotopes. The formation of volatile tungsten oxides therefore leads to mobilized radioactivity, which is unfavorable.

In all these applications, it is crucial to understand the reactivity and stability of the different tungsten oxides. This study aims to investigate the stability of thin layers of ceramic tungsten trioxide as well as a thin layer of its metallic counter part, tungsten dioxide. In this study, we use the highly-oxophilic beryllium to probe the thermal stability of tungsten oxide surfaces. To do so, we evaporate thin layers of beryllium on the two oxide samples and investigate their composition by X-ray photoelectron spectroscopy (XPS).

XPS is chosen because high-resolution XPS not only reveals the elemental composition, but also shows their chemical binding state. This allows for evaluation of the chemical composition and reactions. The temperature is increased in $100\,K$ steps up to a final temperature of $1273\,K$. After each step, the composition is investigated. This approach allows us to make predictions about the thermochemical stability of the compounds. The temperature reactivity study is conducted in situ under UHV (ultra-high vacuum) conditions to avoid the influence of atmospheric gases on the results.

## 2. Experimental

All XPS-measurements are carried out with a *PHI 5600 ESCA* X-ray photoelectron spectroscopy system equipped with a monochromatic Al $K_\alpha$ X-ray source and a hemispherical energy analyzer. The analysis chamber is directly connected to a preparation chamber with an electron beam evaporator equipped with beryllium and a sample heater. Base pressures in both chambers is in the range of $1 \times 10^{-10}$ hPa. For the complete experimental setup, see References [8–10].

The tungsten trioxide is synthesized in an external oven under air. A tungsten plate ($1\,cm^2$) is heated for one hour at $773\,K$. The XP-spectrum shows signal contributions of 88.2% $WO_3$ and 11.8% of its surface compound $WO_2(OH)_2$.

As a precursor for the tungsten dioxide, a tungsten plate of the same size is heated in the same oven flushed with nitrogen at $1000\,hPa$ overpressure under air. The educt is heated for $15\,min$ at $673\,K$, resulting in a thin $WO_3$-layer. The precursor is introduced into the analysis chamber. It is converted into $WO_2$ in a comproportionation reaction at $973\,K$ for one hour. The XP-spectrum shows a signal contribution of 41.41% $WO_2$. The rest of the signal consists of 30.81% $WO_{3-x}$, 25.44% $WO_3$, and 2.33% W as byproducts. This results in a yield of 27.61% and a selectivity of 28.27% regarding $WO_2$. In the X-ray diffractogram, the only visible crystalline phases are $WO_2$ and W (see Figure 1).

The binding energy scale was calibrated using gold, silver, and copper samples, which are stored in vacuo. Before and after each temperature series, the three core levels—Au $4f_{7/2}$ core level at $84.0\,eV$, the Cu $2p_{3/2}$ core level at $932.7\,eV$, and the Ag $3d_{5/2}$ core level at $368.3\,eV$ [8,11,12]—are measured. By measuring these three binding energies, the accuracy of the measured binding energies and the linearity of the binding energy scale is ensured. During the temperature series, the Au $4f_{7/2}$ core level is measured after each step to guarantee, that the measured binding energies are correct.

The acquired XP-spectra are analyzed with the *MultiPak* software solution [13]. For peaks in the W $4f$-spectra a Doniach–Šunjić-Function [14] is applied after subtracting a linear background. Peaks in the Be $1s$ and the O $1s$ binding energy region are analyzed with a Gauss-Lorentz-Function and a Shirley-type background [15]. The pass energy of the analyzer for the high-resolution spectra is set to $2.95\,eV$ and for the spectra of the valence band region to $29.35\,eV$, resulting in energy resolutions of the hemispherical analyzer of $0.04\,eV$, respectively $0.44\,eV$. Taking the apparatus-function into account, a maximum resolution of $0.1\,eV$ is achievable for the high-resolution spectra.

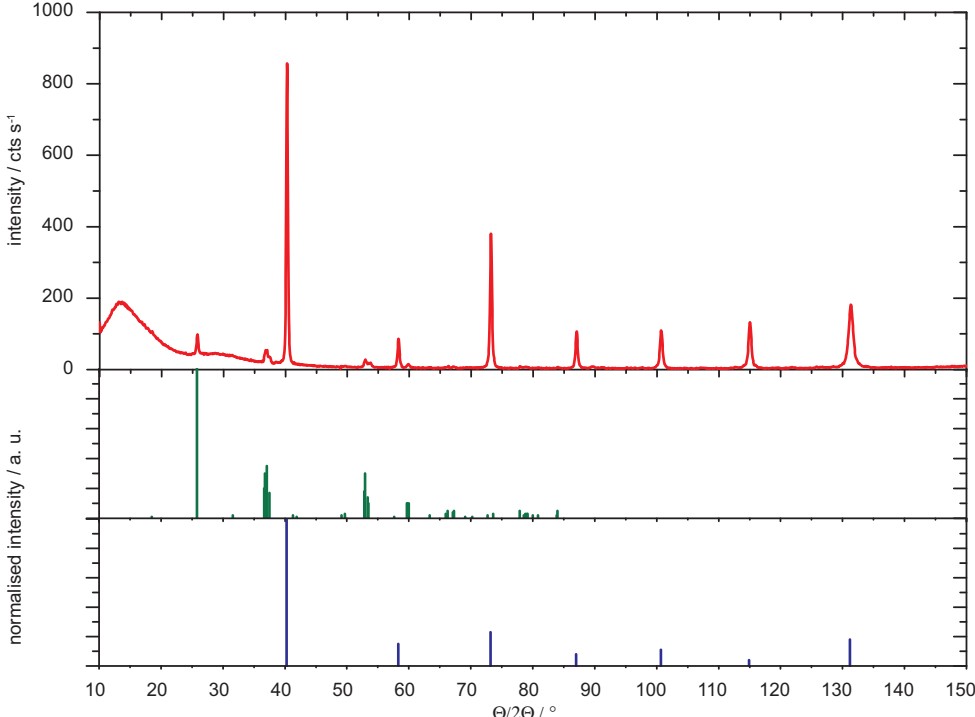

**Figure 1.** X-ray diffractogram of the synthesized $WO_2$-substrate: The diffractogram of the substrate (red) is compared to the reference data of the *PDF*-database of $WO_2$ (green) and W (blue).

## 3. Data Analysis

In this section, the model for the chemical reaction analysis is explained in detail. The model allows for analysis of the chemical reactions at each temperature step. First, a chemical equation, which describes all reactions at the given temperature step, is derived from the spectra. In the following, these equations will be called the main equation. In the final step, the main equation is stripped down to the reaction equations.

Starting with the recorded spectra of the different core levels, the involved compounds can already be identified. If an element is bound in a chemical compound, the signal of this element exhibits a chemical shift. These shifts depend on the chemical surroundings of the element. The different compounds can be identified from these chemical shifts. Unfortunately, the signals of different chemical compounds can overlap. Thus, the spectra must be deconvoluted using fit-functions. The amounts of the specific compounds are directly proportional to the integrals of these fit-functions. Two examples for the deconvolution are shown in Figures 2 and 3.

The signal contribution is heavily influenced by the depth distribution, so only signals originating from the same depth can be compared quantitatively. At the very beginning of the experiment, the samples consist of three layers on top of each other. Beryllium and its oxide form the upper layer. The middle layer is made out of tungsten and its compounds, while the deepest layer is the tungsten substrate, which is beyond the information depth of the used XPS setup. Therefore, we can neglect the substrate for the further analysis. For quantitative analysis, the tungsten oxide layer and the beryllium layer have to be considered.

Only signals originating from one of these layers are used to avoid the influence of depth distribution. For this study, I focus on the spectra of the W $4f$-core level. Since only the relative signal contributions in one core level will be used, it does not matter if the tungsten oxide layer is covered by beryllium at the beginning. The spectra of the W $4f$-core levels reveal only information about the middle layer.

Additionally, tungsten is very mobile, thus, a homogeneous distribution of its compounds within this layer can be assumed [16]. Furthermore, all chemical compounds in this study have a signal contribution in this spectral region, except for one, BeO. So, the W 4*f*-spectra is perfectly suited for the requirements for quantitative analysis.

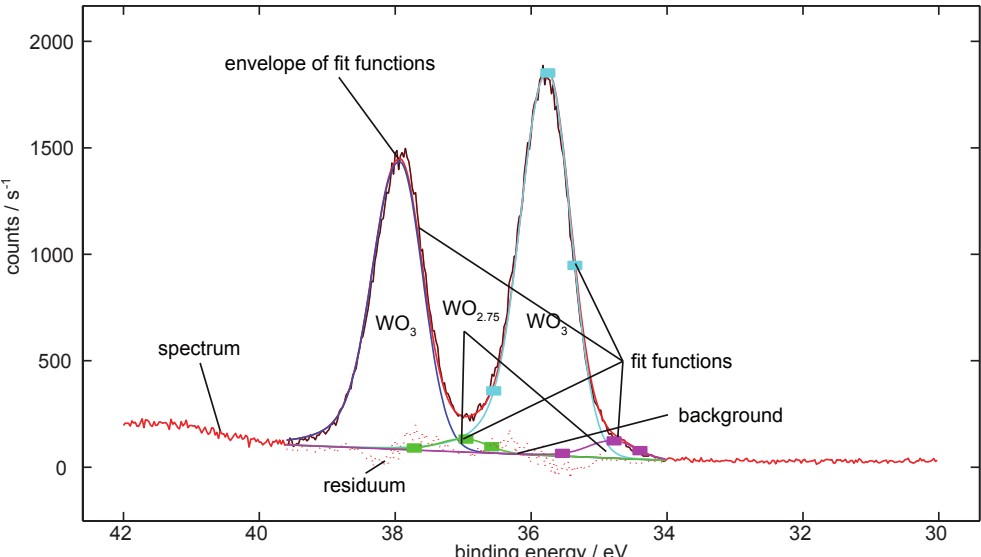

**Figure 2.** A simple example of a spectral deconvolution of a W 4*f*-spectrum of tungsten trioxide with its surface oxide $WO_{2.75}$.

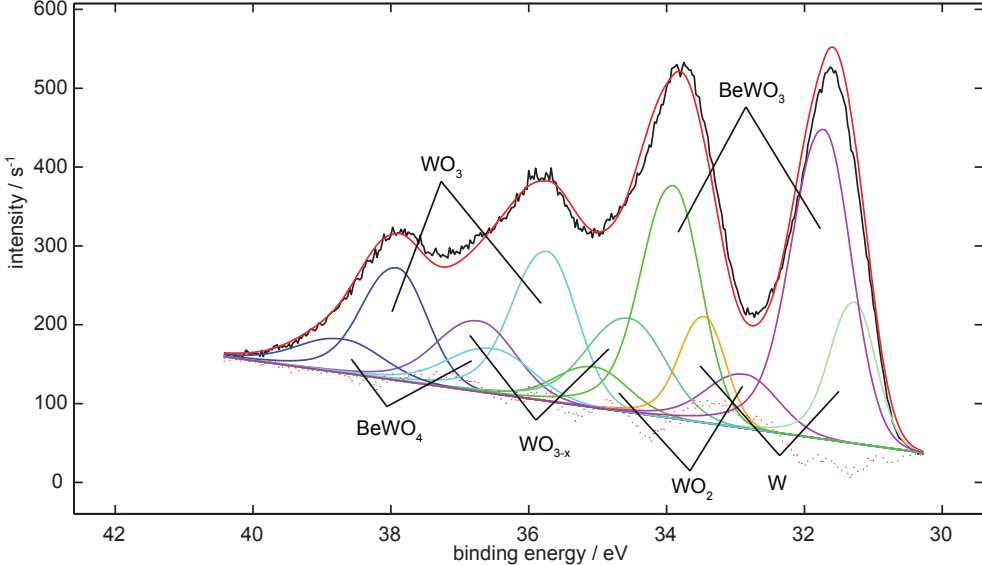

**Figure 3.** A more complex example of the spectral deconvolution procedure. This is the W 4*f*-spectrum of the tungsten trioxide sample after being coated with beryllium.

With this assumption, the intensity of the W 4*f*-spectra only changes with the amount of the compounds and the thickness of the beryllium layer. The beryllium layer attenuates the whole tungsten signal. In order to eliminate the influence of the thickness of the beryllium layer, only the relative

composition of the complete signal is taken into account and not the absolute values. Since only signals from the same core level and the same element are used, even the element-specific cross sections can be neglected.

In addition, the diffusion of beryllium and its oxide does not influence data analysis as long as one can assume that the beryllium compounds are distributed homogeneously throughout the beryllium layer. Even the tungsten oxide layer only gets diluted by a homogeneous contamination of beryllium compounds. This results in a weaker intensity of the signal but not in a change of the fractions of the specific signal contributions.

The spectra of the Be 1$s$ and the O 1$s$ binding energy region provide valuable qualitative information, which is used to verify the results of the analysis of the tungsten spectra.

Only the change of the concentrations of the compounds is important to determine the chemical reaction equations. Therefore, the difference of the signal contribution of the current and the preceding temperature step is calculated. The results are used as stoichiometric factors in the main equation. Negative values mean a decrease of the specific compounds; positive values mean an increase of the specific compounds. With the information from the qualitative analysis of the beryllium spectra the element balance can be agreed. With this step, the main reaction equation is complete.

In the next step, the main reaction equation I is decomposed in the individual reaction equations II, III, .... Therefore, all chemical sensible equations are set up in a system of equations. The result is the main equation:

$$\mathrm{I} = a \cdot \mathrm{II} + b \cdot \mathrm{III} + ... \tag{1}$$

When solving these systems, prefactors $a, b, ...$ for each chemical reaction are obtained. These prefactors are a measure for the reaction rate at the specific temperature. The ratios of the prefactors are the same as the ratios of the rate constants.

In the following, we will show the evaluation of the temperature step at 573 K of the Be-coated $WO_3$-sample in detail as an example of how to extract quantitative information about the chemical reactions from the XPS spectra of the W 4$f$ core level.

At first, the signal contributions from the preceding heating step, here 273 K, have to be subtracted from the signal contributions of the current heating step. As stated above (see Section 3) these values give the alteration of the concentrations of each compound. The result can be written as

$$\underbrace{1.4W + 16.3WO_3}_{17.7W,48.9O} \approx \underbrace{5.4WO_2 + 9.9BeWO_3 + 2.4BeWO_4}_{17.7W,50.1O,12.3Be}. \tag{2}$$

Next, the elemental balances have to be agreed. For this, it is only allowed to consider compounds without tungsten, since Equation (2) is already based on the W 4$f$-signal. In the Be 1$s$-spectrum, Be and BeO can be seen as additional to the tungstate and the bronze (see Figure 5). These compounds are used to agree the balance and we get

$$\mathrm{I:} \quad \underbrace{1.4W + 16.3WO_3 + 1.2BeO + 11.1Be}_{17.7W,50.1O,12.3Be} \longrightarrow \underbrace{5.4WO_2 + 9.9BeWO_3 + 2.4BeWO_4}_{17.7W,50.1O,12.3Be}. \tag{3}$$

Now, the reactions for the system of equation are set up:

$$\text{II:} \quad 2WO_2 + W \quad \longrightarrow 3WO_3, \tag{4}$$

$$\text{III:} \quad WO_3 + Be \quad \longrightarrow WO_2 + BeO, \tag{5}$$

$$\text{IV:} \quad WO_3 + BeO \quad \longrightarrow BeWO_4, \tag{6}$$

$$\text{V:} \quad WO_2 + BeO \quad \longrightarrow BeWO_3, \tag{7}$$

$$\text{VI:} \quad WO_3 + Be \quad \longrightarrow BeWO_3. \tag{8}$$

As shown in Equation (1), the overall equation is now decomposed:

$$I = a \cdot II + b \cdot III + c \cdot IV + d \cdot V + e \cdot VI. \tag{9}$$

After solving Equation (9), the following prefactors are achieved:

$$a = 1.4 \quad b = 1.2 \quad c = 2.4 \quad d = 0.0 \quad e = 9.9.$$

The obtained prefactors comply with the weighting of each single reaction at the current heating step. While reaction V does not take place, the formation of beryllium tungsten bronze is by far the most important reaction at this temperature.

This procedure is used for the evaluation of every temperature step of the experiment. The model is a good tool to determine reactions that take place and find the predominant reaction.

For the sake of clarity, only the results will be described in the following section. The experimental procedure and the subsequent data analysis can be summarized as follows:

1. Heating the sample to the desired temperature;
2. Recording high resolution spectra of the Be $1s$, O $1s$, and W $4f$ and valence band spectra;
3. Spectral deconvolution of the spectra to identify and quantify elements and compounds;
4. Compare the results to the previous temperature step and set up the main chemical equation;
5. Identify possible chemical reactions;
6. Solve the linear equation system to obtain the prefactors for the single reaction equations.

## 4. Results

Both oxidic tungsten specimens are coated with a thin beryllium layer by physical vapor deposition. Each sample is heated in 100 K-steps from 573 K to 1273 K. Each heating step takes 30 min. After each heating step, the sample is cooled down and the spectra are taken with sample temperatures below 373 K. Below, the binding energies of the W $4f_{7/2}$-peak are given. The corresponding W $4f_{5/2}$-peak is at 2.18 eV higher binding energies. For an example of the fitting procedure, please see Figures 2 and 3.

*4.1. Be on WO$_3$*

Analysis of the Spectra

In this section, the analysis of the spectra of the Be-coated tungsten trioxide sample is described. The core level spectra of the W $4f$-region are shown in Figure 4, the Be $1s$-spectra in Figure 5, the O $1s$-spectra in Figure 6, and the valence band spectra in Figures 7. The results of the spectral deconvolution are discussed in detail. All results of the deconvolution of the W $4f$ spectra are compiled in Table 1. A graphical representation of the sample composition at each temperature step is shown in Figure 8. In the following, the spectra analysis is described in detail and the results from the quantitative analysis are given.

Six different compounds can be identified in the W $4f$-spectrum (see Figure 4) of the Be-coated $WO_3$-sample at r. t. The peaks at 32.9 eV, 34.5 eV, and 35.7 eV originate from the three oxidic species $WO_2$, $WO_{3-x}$, and $WO_3$, respectively [17–25]. The peak of elemental tungsten is visible at a binding energy of 31.3 eV. The two remaining peaks at 31.7 eV and 35.7 eV originate from the tungsten bronze $BeWO_3$ and from the beryllium tungstate $BeWO_4$ [26,27]. The tungsten oxides are immediately reduced by the evaporated beryllium, and the tungstate and the tungsten bronze are also formed. In the Be $1s$-spectrum (see Figure 5), four peaks can be identified. The two peaks with the highest intensity at 118.8 eV and 114.7 eV originate from elemental beryllium and its oxide BeO [21,28–33]. The two smaller peaks at 114.2 eV and 113.3 eV originate from the tungstate and the bronze, respectively. The ratios of the signal intensities of these two compounds in the W $4f$- and the Be $1s$-spectra match. A new peak emerges in the spectrum of the valence shell (see Figure 7) at 2.6 eV, which is assigned to the beryllium compounds. The broad peak at 7.2 eV belonging to $WO_3$ slightly looses intensity as $WO_3$ is consumed.

**Table 1.** Signal contributions obtained from the spectral deconvolution of the W $4f$-spectra for the Be-coated tungsten trioxide specimen at the different temperature $T$ steps.

| $T$ [K] | Signal Contributions of the W $4f$-Region [%] | | | | | |
|---|---|---|---|---|---|---|
| | W | $WO_2$ | $WO_{3-x}$ | $WO_3$ | $BeWO_3$ | $BeWO_4$ |
| 293 | 13.0 | 7.8 | 14.8 | 19.6 | 37.2 | 7.6 |
| 573 | 11.6 | 13.2 | 10.5 | 7.5 | 47.2 | 10.0 |
| 673 | 9.4 | 14.7 | 15.3 | 15.0 | 30.6 | 15.1 |
| 773 | 2.6 | 9.4 | 22.3 | 30.8 | 11.9 | 22.9 |
| 873 | 0.0 | 9.8 | 23.4 | 45.3 | 0.0 | 21.6 |
| 973 | 0.0 | 4.7 | 20.2 | 28.6 | 0.0 | 46.5 |
| 1073 | 0.0 | 0.3 | 41.6 | 18.3 | 0.0 | 39.9 |
| 1173 | 0.0 | 0.0 | 43.3 | 34.3 | 0.0 | 22.4 |
| 1273 | 0.0 | 7.3 | 40.8 | 35.2 | 0.0 | 16.7 |

At 573 K, the oxidation of beryllium is completed because no elemental Be is visible anymore. The signal contributions of $WO_3$, $WO_{3-x}$, and tungsten decrease. Out of these three compounds, the tungstates and tungsten dioxide are formed. The signal contribution of $BeWO_3$ reaches its maximum, while the contributions of $WO_3$ and $WO_{3-x}$ reach their minimum. Nearly half of the signal originates from the tungsten bronze. In the valence shell spectrum, the peak at 7.2 eV gets more shallow. The quantitative analysis reveals that the formation of the beryllium tungsten bronze $BeWO_3$ from $WO_3$ is by far the most prominent reaction here.

In the next temperature step, at 673 K, the signal of metallic tungsten decreases even more. The signal contributions of the oxidic tungsten compounds increase. The signal contribution of the beryllium tungstate increases, while the contribution of the tungsten bronze decreases. From the quantitative analysis, we learn that the tungsten bronze $BeWO_3$ is oxidized and decomposes to BeO and $WO_3$ while they partially react further to form beryllium tungstate $BeWO_4$.

At 773 K, the signal contributions of the compounds $WO_2$ and $BeWO_3$ decrease, while all other contributions increase. Here, the oxidation and subsequent decomposition of the tungsten bronze is the most prominent reaction, but also the oxidation of tungsten compounds towards $WO_3$ and $BeWO_4$ takes place.

The contributions of metallic tungsten and $BeWO_3$ vanish at a temperature of 873 K. The signal of $WO_3$ reaches its maximum. The other peak areas nearly stay constant. The oxidation and decomposition of $BeWO_3$ is further ongoing.

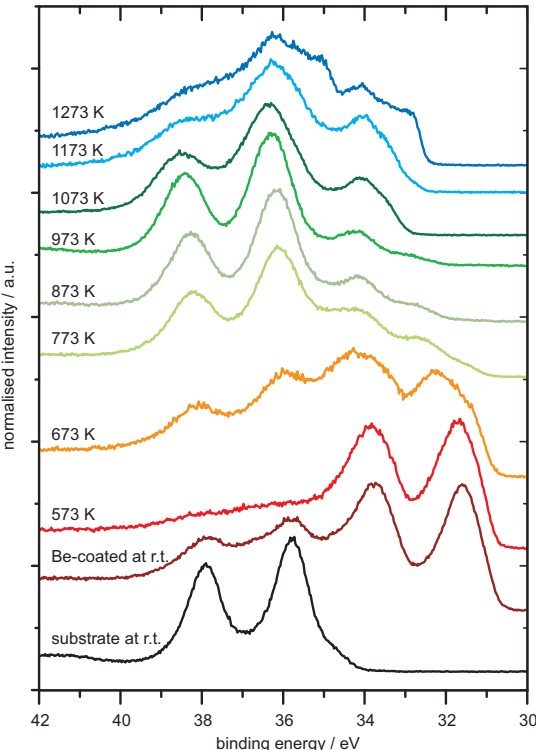

**Figure 4.** W 4*f*-spectra of the coated tungsten trioxide sample.

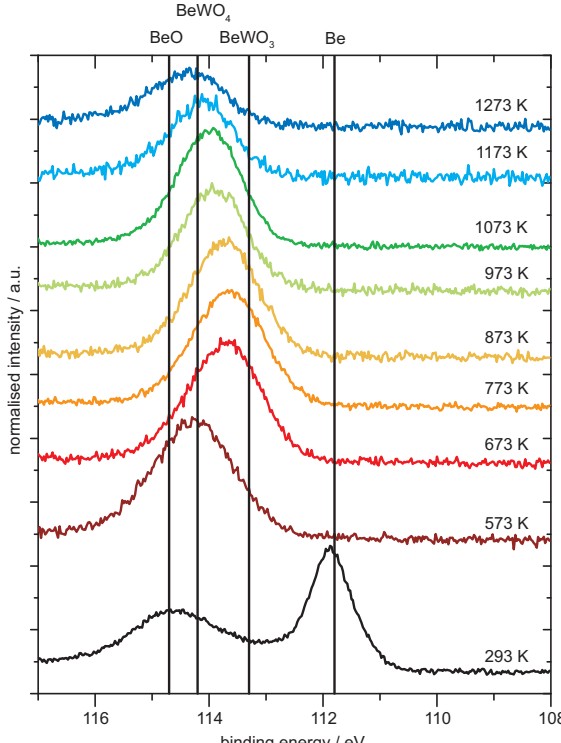

**Figure 5.** Be 1*s*-spectra of the coated tungsten trioxide sample.

At 973 K all oxidic tungsten compounds decrease, while the signal contribution of $BeWO_4$ nearly doubles from 24.9% to 46.5% and reaches its maximum. This coincides with the reaction analysis since the formation of the tungstate is the most prominent reaction. The only other two reactions are the oxidation of $WO_2$ to $WO_3$ at this temperature step.

At 1073 K, the substoichiometric oxides increase their fraction from 21.5% to 41.6%, while $WO_3$ reaches its local minimum of 18.5%. The signal contribution of the tungstate drops by 6.6%. In the spectrum of the valence shell, a new peak evolves at 0.4 eV. This peak is assigned to the W 5*d*-Orbitals of tungsten [17]. In the spectrum of the pure substrate, this peak has a much lower intensity. A small part of the increase is caused by the higher order of the tungsten trioxide and the substoichiometric oxide, but most of the gain is assigned to the tungstate.

Tungsten dioxide completely vanishes at a temperature of 1173 K. The decomposition of $BeWO_4$ proceeds. Accordingly, the contributions of the oxides $WO_3$ and $WO_{3-x}$ increase.

In the final temperature step at 1273 K, the contribution of $BeWO_4$ further shrinks. $WO_2$ is formed again and has a share of the overall signal of 7.3%. Here, $WO_2$ is only a decay intermediate, since, at 1273 K, it is not stable. The substoichiometric oxides slightly decrease while the contribution of $WO_3$ slightly increases. In the spectrum of the valence shell, a new peak evolves at 2.8 eV, which is assigned to $WO_2$.

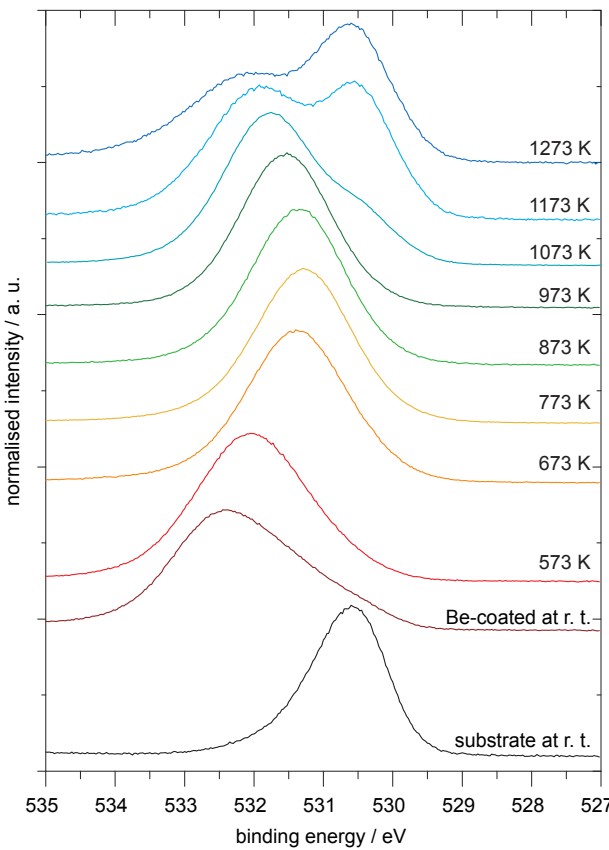

**Figure 6.** O 1*s*-spectra of the coated tungsten trioxide sample.

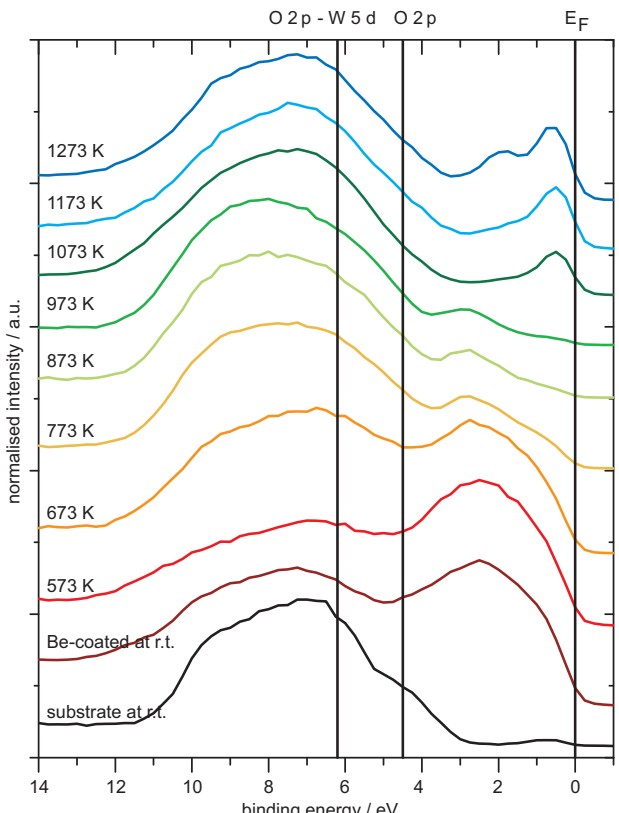

**Figure 7.** Valence band spectra of the coated tungsten trioxide sample.

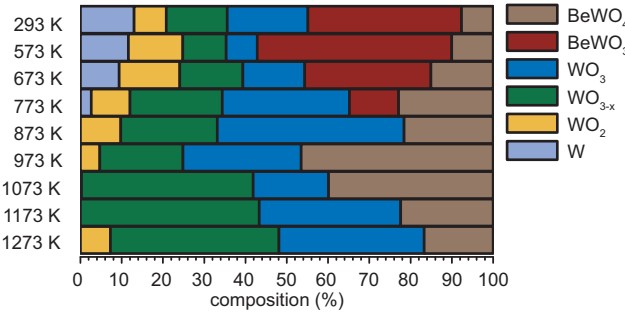

**Figure 8.** Composition of the Be-coated tungsten trioxide sample.

## 4.2. Be on WO₂

Analysis of the Spectra

In this section, the analysis of the spectra and reaction analysis of the tungsten-dioxide-based sample is described. The core level spectra of the W 4*f*-region are shown in Figure 9, the Be 1*s*-spectra in Figure 10, the O 1*s*–spectra in Figure 11, and the valence band spectra in Figure 12. All results of the deconvolution of the W 4*f*-spectra of the WO₂-sample are compiled in Table 2. A graphical representation of the sample

composition at each temperature step is shown in Figure 13. The spectra analysis is described in detail in the following section.

**Table 2.** Signal contributions obtained from the spectral deconvolution of the W $4f$-spectra for the Be-coated tungsten dioxide specimen at the different temperature steps.

| Temperature [K] | Signal Contributions of the W $4f$-Region [%] | | | | | |
|---|---|---|---|---|---|---|
| | W | $WO_2$ | $WO_{3-x}$ | $WO_3$ | $BeWO_3$ | $BeWO_4$ |
| 293 | 28.5 | 25.6 | 20.8 | 4.5 | 16.0 | 4.7 |
| 573 | 22.7 | 21.0 | 20.1 | 4.8 | 27.8 | 3.7 |
| 673 | 21.0 | 30.7 | 16.2 | 5.7 | 21.9 | 4.5 |
| 773 | 21.5 | 31.0 | 17.2 | 5.2 | 20.1 | 5.0 |
| 873 | 17.8 | 32.5 | 16.0 | 7.2 | 21.5 | 5.0 |
| 973 | 18.5 | 33.8 | 15.5 | 8.3 | 15.1 | 8.8 |
| 1073 | 9.8 | 36.7 | 29.9 | 6.9 | 5.8 | 10.9 |
| 1173 | 5.8 | 44.4 | 31.4 | 8.2 | 2.4 | 7.8 |
| 1273 | 85.3 | 0.0 | 4.0 | 9.7 | 0.0 | 0.0 |

After coating of the substrate, six compounds can be identified. The peak pairs of the oxidic tungsten compounds are at 32.9 eV ($WO_2$), 34.7 eV ($WO_{3-x}$), and 35.8 eV ($WO_3$). The peak at 31.4 eV originates from metallic tungsten. The beryllium tungsten bronze has a binding energy of 31.7 eV, and the tungstate has a binding energy of 36.6 eV (see Figure 9). The tungsten dioxide is not reduced by metallic beryllium. In the Be $1s$-spectrum, four compounds can be identified: Beryllium at 111.8 eV; $BeWO_3$ at 113.4 eV; $BeWO_4$ at 114.2 eV; and beryllium oxide at 114.7 eV (see Figure 10). The peak relation of the bronze and the tungstate in the W $4f$- and the Be $1s$-spectrum are the same. In the valence shell region, the intensity of the broad peak between 6.8 eV and 9.2 eV decreases. The clearly defined peak at 0.8 eV assigned to $WO_2$ loses intensity and becomes a shoulder of the peak at 2.1 eV (see Figure 12).

The oxidation of beryllium is finished at a temperature of 573 K. The signal contribution of the bronze increases and reaches its maximum. Accordingly, the peak in the Be $1s$-spectrum shifts to lower binding energies. The signal contributions of all other compounds in the W $4f$-spectrum decrease. This is what we also see in the reaction analysis: The most prominent reaction at this temperature step is the formation of $BeWO_3$.

In the next temperature step, the contribution of $WO_2$ increases by 9.8%. The contributions of $WO_3$ and $BeWO_4$ also increase. The peaks of all other species decrease. The bronze $BeWO_3$ decomposes to BeO and $WO_2$.

At 773 K, the contributions do not change significantly. A closer look with the quantitative model however reveals some reactivity at this step: The tungsten bronze dissociates to $WO_2$ and BeO. W and $WO_3$ is formed by disproportionation. BeO and $WO_3$ form the tungstate.

In the next step, only the signal of elementary tungsten shows a decline. There is no change in the Be $1s$ and the valence shell spectrum. After analysis, we can identify some reactions: At these temperature steps, the tungsten bronze is synthesized via the tungstate as seen above. As there is no metallic beryllium to be seen in the Be $1s$-spectrum, tungsten instead of beryllium acts as the reduction agent here.

At 973 K, the signal contribution of $BeWO_4$ increases while the signal of $BeWO_3$ decreases. The signal of metallic tungsten still decreases. The peaks of the oxidic compounds $WO_2$ and $WO_3$ rise. The contribution of the substoichiometric oxides stay constant. In the other spectra, no change is to be seen. At this step, the tungsten bronze essentially decays or is converted to tungstate.

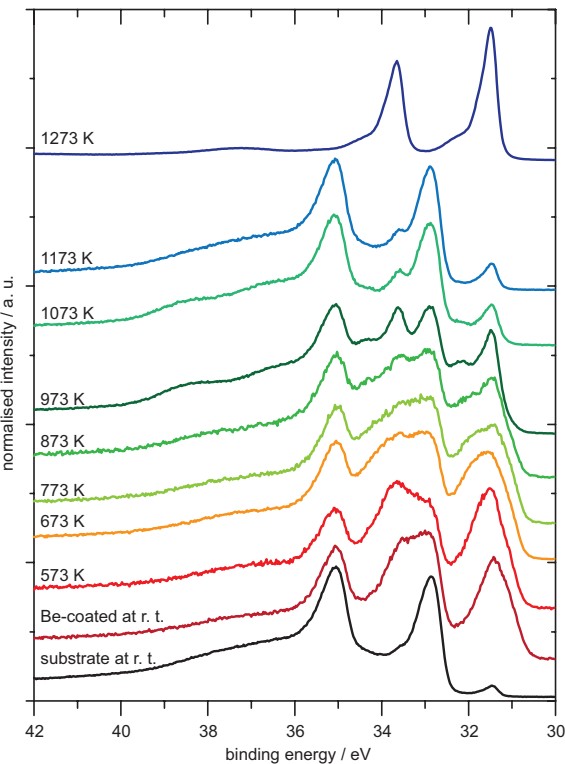

**Figure 9.** W 4*f*-spectra of the coated tungsten dioxide sample.

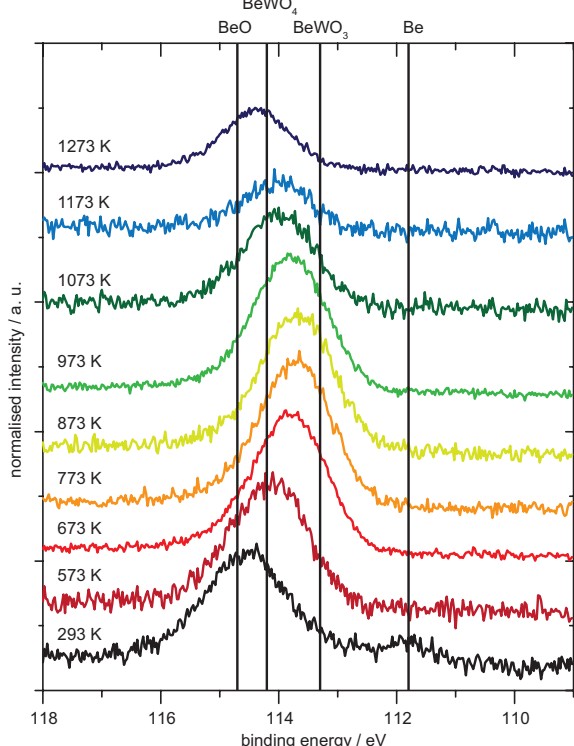

**Figure 10.** Be 1*s*-spectra of the coated tungsten dioxide sample.

The signal contribution of the tungstate reaches its maximum at 1073 K. The share of the signal contribution of the bronze decreases by 9.7%, and the share of the metallic tungsten by 8.7%. The signal contribution of $WO_3$ decreases while the contributions of the other compounds increase. The shoulder in the valence shell spectrum at 0.8 eV becomes a discrete peak.

In the last temperature step, the complete $WO_2$ and the beryllium tungsten compounds are decomposed. The metallic tungsten makes 85.3% of the W $4f$-peak. This is also backed by the reaction analysis. The rest of the signal originates from $WO_3$ and $WO_{3-x}$. In the Be $1s$-spectrum, there are also no beryllium tungsten compounds. The signal of the Be $1s$-region consists of 49.8% beryllium oxide. The remaining Be $1s$-signal is also assigned to beryllium oxide since there are no beryllium tungstate species anymore and no other compounds containing beryllium can be found in this sample.

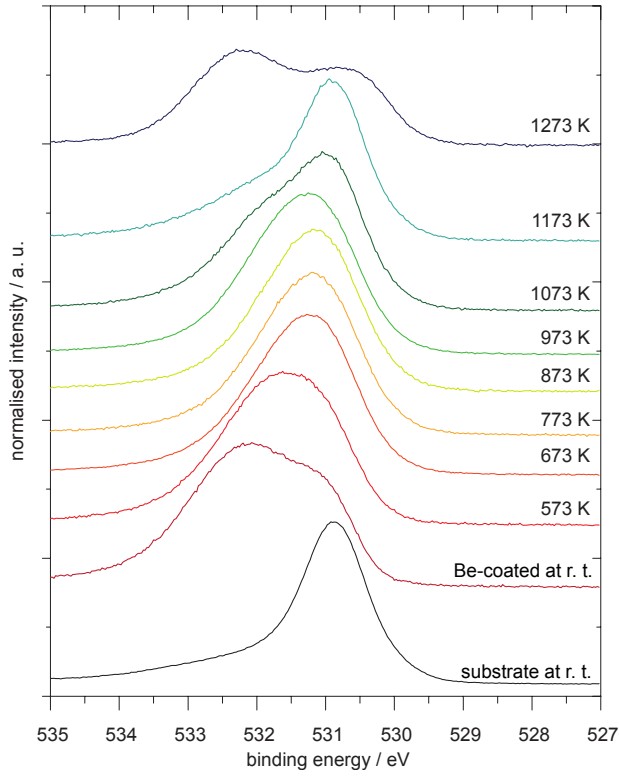

**Figure 11.** O $1s$-spectra of the coated tungsten dioxide sample.

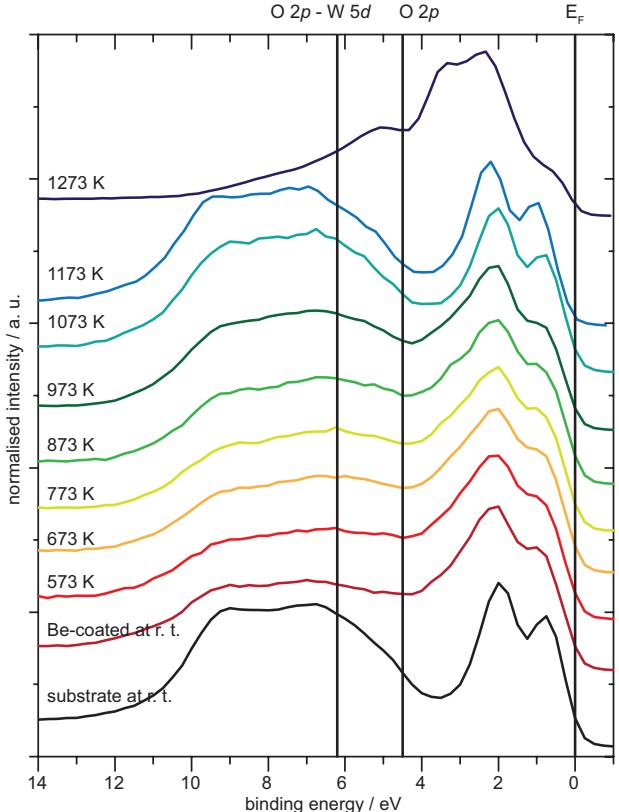

**Figure 12.** Valence band spectra of the coated tungsten dioxide sample.

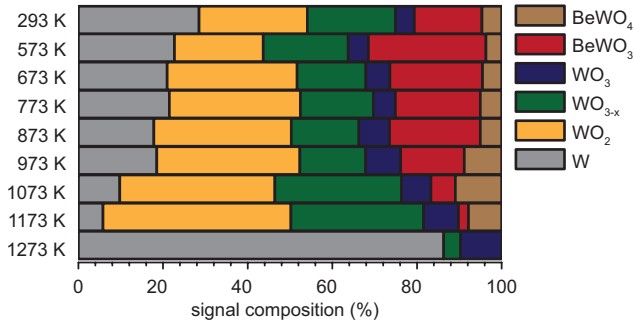

**Figure 13.** Composition of the Be-coated tungsten dioxide sample.

## 5. Discussion

### 5.1. The System Beryllium-Oxygen-Tungsten

In the preceding sections, the reactions of the two investigated ternary systems are determined. To enable these reactions, beryllium and beryllium oxide have to diffuse into the tungsten oxide layer.

Tungsten trioxide and the substoichiometric tungsten oxide show a high diffusivity due to their structure. In these two compounds, tungsten atoms are coordinated octahedrally by oxygen atoms. These coordination octahedrons are linked by their corners. This structure has channels in all three spatial directions through the whole crystal. This structure makes the diffusion possible.

Another point is the structure of tungsten bronze. In this compound, the beryllium ions are embedded in these channels. So, the size of the channels is big enough for the beryllium ions.

On the contrary, tungsten dioxide is surprisingly inert. It shows no reaction until it dissociates at 1273 K. Tungsten dioxide has rutile structure. The rutile structure does not form any channels, so no diffusion into the bulk is possible. Therefore, reactions can only take place at the surface of the crystallites and are negligibly slow.

Nevertheless, in the survey spectra, diffusion can be seen. The diffusion can take place along the grain boundaries.

In the binary system beryllium–tungsten, the formation of alloys takes place [8,28,29]. Here, alloy formation cannot be observed in the ternary system Be-O-W. So, the ternary system shows a completely different chemistry.

However, experiments dedicated to reveal the influence of the different structures of $WO_2$ and $WO_3$ should be the subject of further investigations. X-ray diffraction (XRD) can help to understand the different structures. Experiments using XANES (X-ray absorption near edge structure) spectroscopy will elucidate the evolution and influence of the local geometry of the tungsten atoms [34–38].

### 5.2. The Model

As mentioned above (see Section 3), a homogeneous distribution of the chemically different species is assumed. The model cannot cope with diffusive processes. For the Be-coated tungsten trioxide specimen, an approximately homogeneous distribution can be assumed due to the high mobility of the Be-ions in a $WO_3$-lattice. Additionally, the various chemical reactions in this specimen do their share for a homogenous mixing. Finally, the model delivers results for each temperature step, which are consistent with the observed spectra. For these reasons, the assumption of a homogeneous distribution in the oxide layer is permissible.

Even for the tungsten dioxide layer, a homogeneous distribution can be assumed. At the beginning, the oxide layer of this sample can be divided into two parts—on the one hand, there is the product $WO_2$; on the other hand is the byproduct $WO_3$. For the byproduct, the same is true as for the tungsten trioxide layer. However, the $WO_2$ is nearly inert and does not participate in any chemical reaction until it finally dissociates at a temperature of 1273 K. It can be assumed that the distribution of $WO_2$ at the beginning is homogeneous. Since there is actually no change in the tungsten dioxide, the distribution can be considered as static throughout the experiment until it dissociates.

The results achieved by this model are all chemically sensible, so this is also an indication for its validity.

One of the model's weaknesses is that there is no proof of the uniqueness of the obtained solution. However, the number of results can be decreased significantly by using the information from the Be 1*s*-spectra and the survey-spectra.

The presented model can be improved by taking into account the different information depths throughout the experiment. For example, the information depth of a $WO_3$ sample is higher than the information depth of a pure elemental tungsten. Thus, the information depth changes. The information depth is directly dependent on the inelastic mean free path of photoelectrons (IMFP), which is a function of the sample composition.

In conclusion, the model is well suited to gain chemical information from the XPS experiments and is a valuable tool for the investigation of highly complex systems and for identification of the predominant chemical reactions.

**Funding:** This research received no external funding.

**Acknowledgments:** All experiments are conducted at the Max-Planck-Institut für Plasmaphysik, Boltzmannstr. 2, 85748 Garching, Germany. This study is an excerpt of the master thesis "Investigation of the Ternary System Tungsten-Oxygen-Beryllium" [39] by Martin Köppen at the Technical University of Munich, Department of Chemistry, 85748 Garching, Germany. The author gratefully thanks Michael Fußeder and Florian Kost for the introduction to the XPS experimental setup and for their help in the laboratory; Christian Linsmeier for hosting me in his group; Ulrich Heiz for academic supervision; Stefan Elgeti for his help with the XRD measurements; and Maren Hellwig for proof-reading.

**Conflicts of Interest:** The author declares no conflict of interest.

## Abbreviations

The following abbreviations are used in this manuscript:

r. t.　　room temperature
UHV　　ultra-high vacuum
XPS　　x-ray photoelectron spectroscopy
XRD　　x-ray diffraction

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
