# Peer review of "Comparative Study of the Reactivity of the Tungsten Oxides WO2 and WO3 with Beryllium at Temperatures up to 1273 K"

_condensedmatter, doi:10.3390/condmat4030082_

Round 1

Reviewer 1 Report

The research here presented focuses on the stability of tungsten oxides layers against the reactions catalysed by a Be coating film.  XPS only is used to follow the reaction path and an attempt of data modelling based on temperature depended stochiometric balance is also performed. My general appraisal on this manuscript is that the scientific content is out from the scope of the Condensed Matter journal being the subject of the this investigation a typical surface chemistry research. Moreover, the technical content is not sufficient for the publication. Indeed information coming from the cross comparisons with other experimental  techniques is missing. The minimal characterization content should also include the XRD analysis of the samples (only the as prepared sample is studied by XDR). Moreover, the microstructural changes induced by the films’ reactions are not evidenced, since no (transmission or scanning) electron microscopy study is performed. No information on the thicknesses of the films is provided. Considering the manuscript form: The rationale of the research plan is not clearly discussed; indeed, it is difficult to understand why the Be activation of the reactions is important for sensors or turbines or reactors. In conclusion, I do not suggest the publication of this research in Condensed Matter, whilst a strongly modified study could be submitted in a surface science / coatings journal.          

Author Response

Dear Reviewer,

thank you for your review of my article. Most part of the sample preparation and the complete temperature program is made in-situ in an UHV XPS system. For this reason I only have access to XPS for investigation without exposing the sample to air. In this particular study, I wanted to learn about the influence and reactions of the highly oxophilic beryllium on the two different tungsten oxides WO2 and WO3. Beryllium is chosen because it is a very oxophilic material. As such it is perfect to study the stability of the two different tungsten oxides. I added some lines to the introduction and hope this helps to understand the aim of this study better.
For further investigations of the stability of the two different tungsten oxides and their reactions, experiments with different methods should be conducted e.g. XANES, XRD or SEM/TEM or maybe even angle-resolved XPS techniques. However, aim of this study was to show the very different behaviour and stability of two different tungsten oxide surfaces with a highly oxophilic species, which it did in my humble opinion. Of course, this can and must be the starting point for a wide variety of future experiments. I added some ideas to the discussion section of this article.

Kind regards,

Martin

Reviewer 2 Report

The article "Comparative Study of The Reactivity Of The Tungsten Oxides WO2 And WO3 With Beryllium at Temperatures Up To 1273K" shows the systematic study of the systems and the evolution of the alloys with temperature. The main experimental technique used in this study is X-Ray Photoelectron Spectroscopy. My main remarks in this phase of the review are:

-in the "Experimental" section there should be specified the nominal thickness of the Be film, and also the introductory remarks from the 4.Results section; Starting with the omission of this important information, there are also other important information lacking e.g. the discussion about the three layers in the paragraph 78-85 justifying the nonessential nature of O spectra; 

-also, in the Experimental section the author needs to mention if the binding energies were corrected, to a known binding energy used as a standard, since the author mentions in the line 72 "every core level shows chemical shift"; if there is no such correction I would suggest the author might use as a standard the C-C line from C 1s at 284.6 eV [Applications of Surface Science 1 (1978) 503-514] in the case the Survey spectra were acquired for each step of the reaction, or otherwise, use as a standard the binding energy of one known compound present in all spectra (but this alternative is to be chosen after deconvoluting the spectra);

-the author mentions the "deconvolutions", but none of the spectra shows this kind of analysis. I think they are important, since all the work in the article is concentrated on quantitative results. I also suggest the O 1s spectra to be presented along with the Be 1s and W 4f ones;

-In the "Data Analysis" section the narrative is complicated and the information is not necessary correct, so I suggest the author rewrites this section only keeping the information that is truly important; 

- the XPS signal is strongly dependent on the inelastic mean free path of each species so I suggest the author takes this into consideration while reconstructing the narrative in the "Data analysis" section;

-there are some typos like the using the lowercase letters while mentioning a figure or a table and using alternatively the abbreviation fig. and the full word figure, the abbreviation of the word see.

Starting from these corrections, additional ones might be necessary for the calculations of stoichiometries.

Author Response

Dear Reviewer,

thank you for your review, detailed remarks and recommendations. In the following, I will comment on your remarks:

1. You are completely right, it would be better to include the thicknesses of the Be layers, unfortunately I haven’t measured them back in the days, when I made the experiments. I think it is better to omit this information, then to give any rough and maybe wrong estimates. I rewrote the paragraphs line 88 - 102, I hope, it is more clear now.

2. The binding energy scale was carefully calibrated with the core levels of Au, Cu and Ag. You can find this information in the „Experimental“ section. We cannot use the C1s signal, because there is simply no carbon contamination on our sample. However, at the beginning of each experimental series, we calibrated the energy scale using the copper, gold and silver samples. After each experimental step, the binding energies are verified using a gold sample. I explained my calibration procedure in more detail. line (61-67) I could not use the C 1s signal, because we simply don’t have any carbon contamination.
I rewrote the passage with „every core level shows chemical shift“ since it can be easily be misunderstood.

3. O1s spectra are added (Figs. 6 & 11). Example graphs for the deconvolution of the spectra are added (Figs 2 & 3)

4. I decided to explain my mode of evaluation on one example for one temperature and removed the detailed analysis of the other temperature steps. I tried to simplify to analysis section. Please let me know, what you think about this shorter version.

5. I added a paragraph to the discussion section, explaining how the model can be improved by taking into account variations of the IMFP and information depth.

Kind regards,

Martin

Round 2

Reviewer 2 Report

Dear author,

Please check again the Briggs and Grant book and see there is a difference between the instrument calibration (which needs the measurement of Au, Cu and/or Ag samples and it needs to be done after major changes in the electronics of the instrument) and the binding energy calibration for isolating/ semiconducting samples.

Best regards!